# Isolation and Characterization of Fungal Endophytes from *Petiveria alliacea* and Their Antimicrobial Activities in South Florida



Ganesh Khadka [1], Thirunavukkarasu Annamalai [2], Kateel G. Shetty [1], Yuk-Ching Tse-Dinh [2] and Krish Jayachandran [1,*]

[1] Department of Earth and Environment, Florida International University, Miami, FL 33199, USA; gkhad001@fiu.edu (G.K.); shettyk@fiu.edu (K.G.S.)
[2] Department of Chemistry and Biochemistry, Florida International University, Miami, FL 33199, USA; athiruna@fiu.edu (T.A.); ytsedinh@fiu.edu (Y.-C.T.-D.)
[*] Correspondence: jayachan@fiu.edu; Tel.: +1-305-348-6553

**Abstract:** Microorganisms associated with medicinal plants are of great interest as they are the producers of important bioactive compounds effective against common and drug-resistant pathogens. The characterization and biodiversity of fungal endophytes of the *Petiveria alliacea* plant and their antimicrobial production potential are of great interest as they are known for their antimicrobial and anticancer properties. In this study, we investigated the endophytic fungal microbiome associated with *P. alliacea*, and the endophytic fungal isolates were classified into 30 morphotypes based on their cultural and morphological characteristics. Ethyl acetate extract of fungal endophytes was obtained by liquid–liquid partitioning of culture broth followed by evaporation. The crude extract dissolved in dimethyl sulfoxide was screened for antimicrobial activity against three bacterial strains (*Escherichia coli* ATTC 25902, *Staphylococcus aureus* ATTC 14775, *Bacillus subtilis* NRRL 5109) and two fungal strains (*Candida albicans* ATTC 10231 and *Aspergillus fumigatus* NRRL 5109). Among the crude extracts from endophytes isolated from leaves, 65% of them showed antimicrobial activity against the bacteria tested. Similarly, 71 and 88% of the fungal crude extracts from endophytes isolated from root and stem, respectively, showed inhibitory activities against at least one of the bacterial strains tested. Crude extracts (at a concentration of 10 mg/mL) from ten of the fungal isolates have shown a zone of inhibition of more than 12 mm against both Gram-positive and negative bacteria tested. Sequenced data from isolates showing strong inhibitory activity revealed that *Fusarium solani*, *F. proliferatum*, and *Fusarium oxysporium* are the major endophytes responsible for bioactive potential. These results indicate that *Petiveria alliacea* harbors fungal endophytes capable of producing antimicrobial metabolites. Future studies need to focus on testing against drug-resistant bacteria (ESKAPE group) and other pathogenic bacteria and fungi.

**Keywords:** endophytes; antimicrobials; medicinal plants; fungi



## 1. Introduction

Since ancient times, medicinal plants have been used as a primary source of medicine [1,2]. Even in these modern times, plant remedies are still the most important therapeutics to treat diseases [3]. According to the survey report of the World Health Organization, "Almost 80% of the world's population from developing countries still rely on traditional medicine that involves the use of plant extract for their primary health care" [4,5]. Endophytes are an important component of the plant microbiome and are known to be highly abundant and diverse.

Endophytic fungi live asymptomatically within the host plant tissues [6,7]. They can be found in a wide range of host plants, and there is at least one endophyte in all the

plants studied so far, and hundreds or thousands of endophytic fungus species may be present [7,8].

Endophytes are known to produce a multitude of secondary metabolites and are considered a natural reservoir of novel bioactive compounds of medical importance [9,10]. Currently, there is a growing interest in the bioprospecting of endophytes for bioactive compounds and their potential application in biotechnology and biomedicine.

Endophytic fungi produce some of the widely used antibiotic and cancer treatment drugs. Some examples of these antibiotics and anticancer drugs include Penicillenols extracted from *Penicillium* species and Taxol extracted from *Taxomyces andreanae*, which are the most effective and popular cancer drugs extracted from endophytic fungi. Similarly, Clavatol from *Torreya mairei*, Sordaricin from *Fusarium* spp. Jeserone from *Pestalotipsis jester* and Javanicin from *Chloridium* spp. are well known for possessing strong antifungal and antibacterial properties against several infectious disease-causing agents. Pestacin was isolated from *Pestalotiopsis microspora* and also possesses strong antioxidant properties [11]. This evidence suggests that endophytic fungi provide a promising source of novel drugs.

As they are an important source of novel antibiotics, only about a tenth of the estimated one million terrestrial fungal endophytes have been studied, and a lot more remain unknown [12]. The spread of antibiotic-resistant bacteria is a growing problem worldwide; it is critical to find new sources of antimicrobials to address this problem. Recently, there has been a significant interest in endophyte research due to their immense potential as a source of novel bioactive compounds. The choice of the plant to be used for exploring endophytes for alternative sources of bioactive compounds is important [13]. Therefore, medicinal plants are a valuable source for bioprospecting endophytes. Traditional ethnobotanical knowledge may assist in narrowing down the plants as targets for investigating the production of novel antimicrobial compounds. However, in certain cases, plant availability for commercial production can be a limiting factor. Many bioactive compounds in medicinal plants are produced by their endophytes. Hence, it is more appropriate to explore the endophytes associated with those medicinal plants [13].

South Florida also plays a crucial part in the ethnobotanical sector of US history. Tropical and sub-tropical medicinal plants present in South Florida provided an important source of remedy for Native Americans and early settlers in Florida [14]. Different varieties of medicinal plants have adapted to grow in this climatic condition. In addition to that, native people who live in South Florida, such as the Seminole and Mikasuki tribes, still use traditional herbs passed down by their ancestors. Scientists have started researching medicinal plants for therapeutic purposes, and they have been successful in finding some important chemicals from South Florida's native medicinal plants. These include chemicals from Saw Palmetto fruits to reduce swelling associated with prostate cancer [15].

Although the potential to discover new antimicrobial compounds still exists in South Florida, the rapid loss of natural habitats and biodiversity of medicinal plants threaten the chances of discovering novel antimicrobial products from these plants. So, immediate screening of medicinal plants for their valuable compounds from endophytes is needed to fight against emerging pathogens and would be an important reason to conserve these valuable plant resources in the region.

*Petiveria alliacea*, which is also known by its common name Guinea Hen Weed, belongs to the family Phytolaccaceace, which is the most primitive family in the order of Caryophyllales [16]. This family has 17 genera and 120 tropical and sub-tropical species that are found across South and North America. Herbs, shrubs, and, on rare occasions, trees with little flowers and alternating leaves are included in this family [17,18]. Research conducted on plant part extracts has demonstrated important information about its bioactive potential. *Petiveria alliacea* leaves contain important chemicals such as antifungal, antiviral, and anti-inflammatory compounds [19]. It is used as an antirheumatic, a soothing agent, and for restorative purposes [20,21].

The endophytic fungal communities of *Petiveria alliacea* from South Florida and their bioactive potential have not been examined before. The purpose of this study was to inves-

tigate the endophytic fungal microbiomes of the native medicinal plant *Petiveria alliacea* in South Florida as a source of novel antimicrobials to combat the emerging new biothreat and antibiotic resistance problem.

## 2. Materials and Methods

### 2.1. Collection of Host Medicinal Plants

The healthy plant samples of *Pativeria alliacea* were collected from Possum Trot Tropical Fruit Nursery (25°32′7″ N; 80°28′37″ W) located in Miami's Redland Agricultural district, Florida. Samples of fresh, healthy leaves, stems, and roots of *Pativeria alliacea* were collected in separate zipper lock bags, labeled, stored in a cooler on ice, transported to the laboratory, and stored at 4 °C until processing as described previously by Tan et al., 2018 [22]. Samples were processed within 24 h of collection. Specimens were identified by Robert L. Burnum from Possum Trot farm, who is a taxonomist in native medicinal plants in South Florida.

### 2.2. Isolation of Endophytic Fungi

Isolation and cultivation of endophytic fungi were carried out from the leaf, stem, and root of the sampled plants (Figure 1). Surface sterilization of the plant's parts was performed by a modified protocol from Tan et al., 2018. For surface sterilization, the segments were soaked in 70% ethanol for 30 s, then sodium hypochlorite (2.5% available chlorine) for 5 min, followed by three successive rinses with sterile distilled water and blotted dry using sterile filter paper. Sterile segments were immediately aseptically cut into smaller fragments (0.5 cm × 0.5 cm), and individual sample fragments were placed horizontally on potato dextrose agar (PDA) supplemented with chloramphenicol (200 µg/mL) and streptomycin sulfate (100 µg/mL). Nine stem/leaf/root fragments per plant sample were seeded onto agar plates. The effectiveness of the surface sterilization was confirmed by the absence of microbial growth on the triplicate PDA plates inoculated with 100 µL aliquots of the final rinse water and on the triplicate PDA plates with a surface imprint of surface sterilized sample segment. Plates were incubated at room temperature (28 °C) for more than a week. Inoculated plates were observed every day for fungal growth from plant tissue. Fungal hyphal tips from the growth were transferred onto the fresh PDA, and the sub-culturing of the fungal isolate was continued until a pure culture was obtained. Observations on the morphological characteristics (colony color, surface, margin, pattern, etc.) of the fungal isolates were recorded. Cultures of individual endophytic fungal isolates were stored in 2.0 mL cryovials containing 15% glycerol in PDB medium at −80 °C.

### 2.3. Crude Extract Preparation for Secondary Metabolites

Individual endophytic fungal isolates from *Petiveria alliacea* were grown on PDA medium for 7 days at 28 °C. After that, three discs (5 mm) of each isolate were picked and inoculated individually into a 250 mL Erlenmeyer flask containing 80 mL of potato dextrose broth (PDB) and incubated at 28 °C for 60 days in an orbital shaker set at 125 rpm. Endophytic extract preparation was carried out following a modified protocol from Pansanit and Pripdeevech, 2018 and Sharma et al., 2016 [23,24]. The fungal broth was centrifuged for 5 min at 4000× *g* to obtain a cell-free supernatant (CFS). The supernatant was decanted into a separate vial, and the CFS and the pellet were extracted using 50 mL and 30 mL of ethyl acetate, respectively, at room temperature for 3 days. The resulting decanted organic phase from CFS and pellet were combined, filtered, and concentrated by using a Buchi R-300 vacuum rotary evaporator (BUCHI Corporation, New Castle, DE, USA) at reduced pressure at 45 °C, resulting in 10 mL of fungal crude extract of endophyte. Later, the solvent from the crude extract was further evaporated using a vacuumed rotary evaporator (Buchi R-300) at 45 °C to yield the crude metabolite. The dry solid was dissolved in dimethyl sulphoxide (DMSO) to obtain a final concentration of 10 mg/mL and stored at 4 °C. The crude extract and DMSO extract were used for the initial and final screening of antimicrobial activity, respectively.

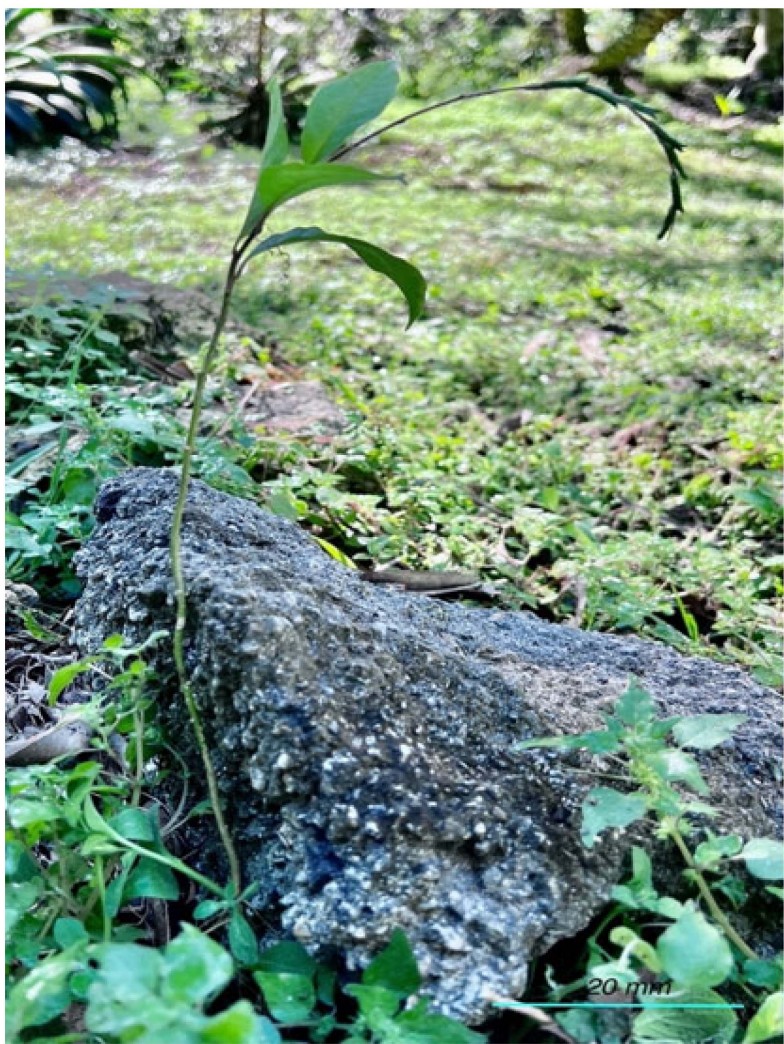

**Figure 1.** Adult plant of *Petiveria alliacea* herb with visible seed. (Bar = 20 mm).

### 2.4. Testing of Antibacterial and Antifungal Activity

The endophytic fungal crude extracts were screened for their antimicrobial activity using the agar diffusion method against Gram-negative (*Escherichia coli* ATTC 25902), Gram-positive (*Staphylococcus aureus* ATTC 14775, *Bacillus subtilis* NRRL 5109), unicellular fungus (*Candida albicans* ATTC 10231), and multicellular filamentous fungus (*Aspergillus fumigatus* NRRL 5109). Tested bacteria and fungi were seeded on Muller–Hinton agar (MHA) and Potato Dextrose Agar (PDA) plates, respectively (Figure 2). On each plate, three equally spaced wells were cut into the surface of agar using a sterile cork-borer (6 mm), and 100, 150, and 200 μL of endophytic fungal crude extract or 10 mg/mL DMSO were then dispensed into three separate wells then incubated at 28 °C [23,24]. Antimicrobial activity was assessed by measuring the inhibition diameter (mm) zones at 24 and 48 h. Ampicillin sodium 100 μg/mL (positive control 1) and Fluconazole 30 μg/mL (positive control 2) were employed as positive controls, while 10% DMSO was used as a negative control. The experiment was performed in triplicates.

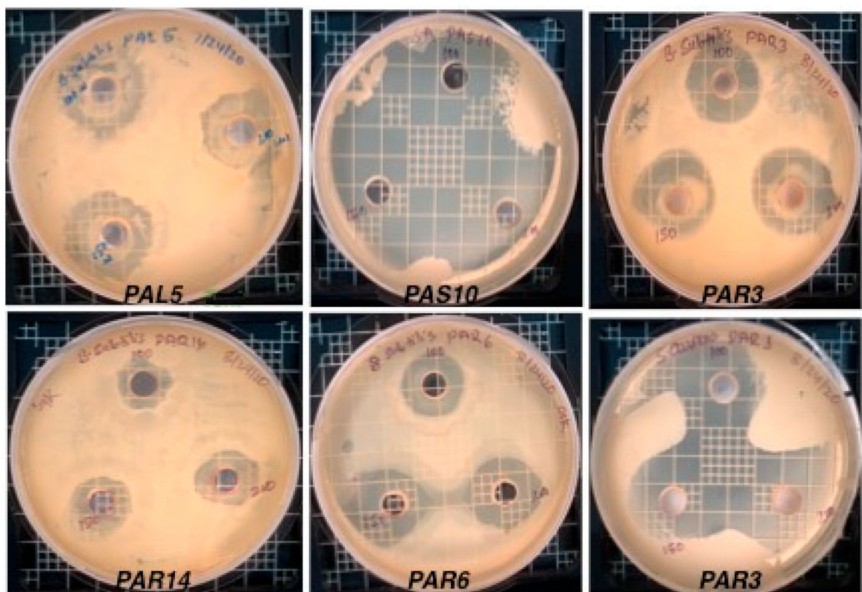

**Figure 2.** Antibacterial activity of the extracts of fungal endophytes against selected bacterial strains.

*2.5. Sequencing of Endophytic Isolates*

The isolates that showed positive antimicrobial activity were identified using molecular methods from Azenta Life Science. Briefly, the colony samples were lysed using a crude NaOH lysis technique to be directly used in PCR. To amplify ribosomal internal transcribed spacers (ITS), the primers ITS1 (5′TCCGTAGGTGAACCTGCGG3′) and ITS4 (5′TCCTCCGCTTATTGATATGC-3′) were employed in a polymerase chain reaction (PCR) assay [25]. Amplified samples were spot-checked using gel electrophoresis to check for robust amplification, along with a negative control to check for contamination. Following amplification, enzymatic cleanup was performed according to Azenta Life Sciences SOP using Exonuclease I—Shrimp Alkaline Phosphatase (ExoSAP), and dye-terminator sequencing was performed by Azenta Life Sciences Inc. (South Plainfield, NJ, USA) using Applied Biosystems BigDye version 3.1. Sequencing-specific primers were used to generate bidirectional reads. The reactions were then run on Applied Biosystem's 3730xl DNA analyzer. Sequences were then compared with the National Center for Biotechnology Information (NCBI) database (http://www.ncbi.nlm.nih.gov), accessed on 5 January 2023, and accession numbers were assigned to each sequence. The assigned accession numbers for each isolate were submitted and published to the NCBI GenBank database. Phylogenetic analysis was performed for isolates showing strong inhibition (>20 mm) based on sequence data from ITS gene using Mega software version 11 using a Neighbor-joining phylogenetic tree. The bootstrap value was chosen to be 100 for the percentage of bootstrap replications supporting the branch [26].

*2.6. Statistical Analysis*

The Shannon–Weiner Index ($H'$) was used to calculate the fungal diversity of endophytic fungi using the formula below.

$$H' = -\Sigma(\text{P}i \times \ln \text{P}i)$$

where P$i$ was calculated as P$i = \frac{ni}{N}$, and $ni$ represents numbers of the individual segment from which fungal endophytic isolates were isolated, and $N$ is the total number of segments incubated [27,28].

Similarly, the Colonization rates (CR%) of the fungal isolates from *Petiveria alliacea* were calculated as follows: CR% = Nsc/Nss × 100 where Nsc represents the number of segments infected by fungal isolates and Nss represents the total numbers of plant segments investigated. The isolation ratio (IR%) of the fungal strains was calculated as follows:

(IR% = Ni/Nt × 100) where Ni is the number of segments from which fungal isolates were isolated, and Nt is the total number of plant segments incubated [29].

$$CR\% = \frac{No.\ of\ infected\ segment}{No.\ of\ segment\ studied} \times 100;\ IR\% = \frac{No.\ of\ segment\ from\ which\ fungi\ isolated}{No.\ of\ segment\ incubated} \times 100$$

The diversity index was analyzed by two-way ANOVA. SPSS version 20.0 was used to perform all statistical analyses (SPSS Inc., Chicago, IL, USA).

## 3. Results

### 3.1. Collection and Pure Culture Isolation

From the 289 sampled plant parts (representing a total of 99 roots, 99 stems, and 99 leaves), a total of 279 isolates were counted and are categorized as 30 morphotypes based on their morphological appearances (Figure 3). Eleven endophytic fungi were morphologically different from each other and isolated from the root sample; 9 were from the stem, and 10 were isolated from the leaf sample (Table 1) [30].

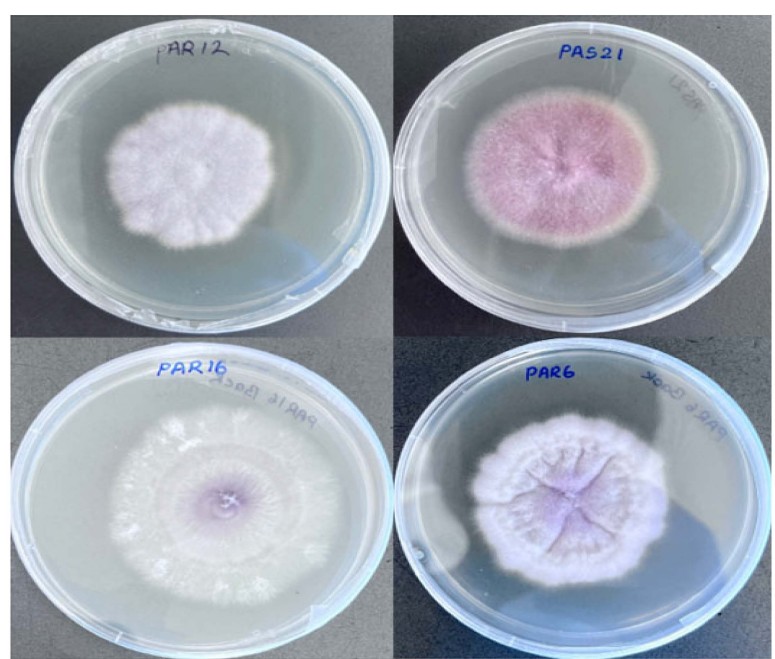

**Figure 3.** Colony morphology of endophytic fungi isolated from *Petiveria alliacea* on potato dextrose agar.

**Table 1.** Endophytic fungal isolates from *Petiveria alliacea*.

| Tissue | Segment Studied | Infected Segments | Total Isolates | *Endophytic* spp. | CR% | IR% | Shanon_*H'* |
|--------|-----------------|-------------------|----------------|-------------------|------|------|-------------|
| Root | 99 | 83 | 86 | 11 | 83 | 86.5 | 2.366 |
| Stem | 99 | 95 | 97 | 9 | 96 | 98 | 2.189 |
| Leaf | 99 | 94 | 96 | 10 | 95 | 97 | 2.282 |

The highest percentage of the colonization rate (97%) was found in the stem sample, and the highest percentage of the isolation rate (98%) was also found in the stem sample. Similarly, the highest Shannon–Weiner diversity index was found in the root sample from *Petiveria alliacea* (Table 1).

### 3.2. Antimicrobial Activity of Crude Fungal Extract Using Agar Well Diffusion Method

The antimicrobial activities of crude extracts from 30 endophytic fungal isolates were evaluated against test organisms. Crude extracts from the majority (83%) of the isolates showed positive antimicrobial activity in the initial screening (Tables 2 and 3). Out of

the fungal endophyte crude extracts that showed antimicrobial activity, 65% of the leaf endophyte extracts, 88% of the stem endophyte extracts, and 71% of the root endophyte extracts showed activity against at least one of the bacterial strains tested. Only 10% of the fungal crude extracts prepared from leaf endophytes demonstrated activity against at least one of the fungal strains tested. None of the crude fungal extracts prepared from the stem and root samples showed activity against the fungal strains tested. The largest zone of inhibition was observed against *S. aureus* (27.8 ± 0.5 mm) inhibited by the PAS-14 isolate, *E. coli* (27 ± 0.1 mm) inhibited by the PAL-23 isolate, and *B. subtilis* (25.8 ± 0.1 mm) inhibited by the PAL-05 isolate crude extracts. Crude extracts from two endophytes, PAL-05 from the leaf and PAR-06 from the root, showed broad-spectrum activity against both bacteria and fungi. Isolate *Fusarium solani* PAL-05 showed a maximum inhibition zone of 25.8 ± 0.1 mm and 4.1 ± 0.2 mm against *B. subtilis* and *A. fumigatus*, respectively. Similarly, the isolate *Fusarium oxysporium* PAR-06 showed a maximum inhibition zone of 20.0 ± 0.2 mm, 22.2 ± 0.1 mm, and 9.8 ± 0.3 mm against *S. aureus, B. subtilis,* and *C. albicans*, respectively. Three of the endophytic isolates, *Fusarium oxysporum* PAR-16, *Fusarium oxysporum* PAS-04, and *Fusarium solani* PAS-13, demonstrated antimicrobial activity against both Gram-negative and Gram-positive bacteria but did not show any activity against the fungal strains tested. None of the isolates examined showed inhibition against all the test organisms.

**Table 2.** Antibacterial activities of the crude extracts of endophytic fungi isolated from *Petiveria alliacea* against selected bacterial pathogens (the numbers are the diameters of the zones of inhibition in mm; mean ± SD; *n* = 3).

| Endophytes | *E. coli* | | | *S. aureus* | | | *B. subtilis* | | |
|---|---|---|---|---|---|---|---|---|---|
| μL | 100 | 150 | 200 | 100 | 150 | 200 | 100 | 150 | 200 |
| PAL 04 | - | - | - | - | - | - | 18.3 ± 0.5 | 22.2 ± 0.3 | 24.0 ± 0.1 |
| PAL 05 | - | - | - | - | - | - | 22.2 ± 0.3 | 23.1 ± 0.4 | 25.8 ± 0.1 |
| PAL 13 | - | - | - | 6.1 ± 0.2 | 9.0 ± 0.2 | 10.0 ± 0.2 | 9.4 ± 0.1 | 10.2 ± 0.1 | 11.3 ± 0.2 |
| PAL 23 | 22.3 ± 0.5 | 24.5 ± 0.2 | 27.0 ± 0.1 | 20.1 ± 0.1 | 23.3 ± 0.2 | 25.7 ± 0.1 | - | - | - |
| PAR 03 | 12.1 ± 0.3 | 14.2 ± 0.1 | 18.1 ± 0.3 | 20.2 ± 0.2 | 22.0 ± 0.3 | 23.9 ± 0.2 | 18.2 ± 0.3 | 20.1 ± 0.1 | 22.1 ± 0.2 |
| PAR 06 | - | - | - | 16.2 ± 0.2 | 18.1 ± 0.3 | 20.0 ± 0.2 | 16.1 ± 0.2 | 19.3 ± 0.2 | 22.2 ± 0.1 |
| PAR 16 | 12.0 ± 0.3 | 14.0 ± 0.1 | 16.2 ± 0.3 | 20.2 ± 0.1 | 22.2 ± 0.3 | 24.1 ± 0.2 | 13.3 ± 0.3 | 16.1 ± 0.1 | 18.0 ± 0.2 |
| PAS 04 | 16.3 ± 0.1 | 18.2 ± 0.3 | 20.1 ± 0.2 | 14.1 ± 0.3 | 16.1 ± 0.1 | 18.2 ± 0.1 | 17.8 ± 0.1 | 20.1 ± 0.2 | 22.2 ± 0.3 |
| PAS 13 | 18.1 ± 0.3 | 20.2 ± 0.3 | 22.0 ± 0.3 | 16.1 ± 0.3 | 17.2 ± 0.3 | 17.5 ± 0.5 | 16.2 ± 0.3 | 18.2 ± 0.2 | 20.0 ± 0.1 |
| PAS 14 | - | - | - | 24.1 ± 0.3 | 25.9 ± 0.4 | 27.8 ± 0.5 | 2.0 ± 0.1 | 3.9 ± 0.1 | 5.8 ± 0.3 |
| PAS 26 | - | - | - | 23.7 ± 0.5 | 26.0 ± 0.3 | 27.5 ± 0.4 | 14.2 ± 0.3 | 16.1 ± 0.3 | 17.5 ± 0.3 |
| Positive control 1 | 15.0 ± 0.1 | 16.00 ± 0.2 | 16.9 ± 0.2 | 16.0 ± 00 | 16.3 ± 0.2 | 16.9 ± 0.2 | 20.0 ± 0.1 | 20.1 ± 0.2 | 20.5 ± 0.2 |
| Positive control 2 | - | - | - | - | - | - | - | - | - |
| Negative control | - | - | - | - | - | - | - | - | - |

**Table 3.** Antifungal activities of the crude extracts of endophytic fungi isolated from *Petiveria alliacea* against selected fungal pathogens (the numbers are the diameters of the zones of inhibition in mm; mean ± SD; *n* = 3); Positive control: Fluconazole (30 μg/mL); Negative control: 10% DMSO.

| Endophytes | *C. albicans* | | | *A. fumigatus* | | |
|---|---|---|---|---|---|---|
| μL | 100 | 150 | 200 | 100 | 150 | 200 |
| PAL 04 | - | - | - | - | - | - |
| PAL 05 | - | - | - | 1.0 ± 0.2 | 2.0 ± 0.2 | 4.1 ± 0.2 |
| PAL 13 | - | - | - | - | - | - |
| PAL 23 | - | - | - | - | - | - |
| PAR 03 | - | - | - | - | - | - |
| PAR 06 | 6.1 ± 0.2 | 8.1 ± 0.1 | 9.8 ± 0.3 | - | - | - |
| PAR 16 | - | - | - | - | - | - |
| PAS 04 | - | - | - | - | - | - |
| PAS 13 | - | - | - | - | - | - |
| PAS 14 | - | - | - | - | - | - |
| PAS 26 | - | - | - | - | - | - |
| Positive control | 24.1 ± 0.3 | 24.1 ± 0.3 | 24.6 ± 0.3 | 17.1 ± 0.3 | 17.4 ± 0.1 | 17.8 ± 0.3 |
| Negative control | - | - | - | - | - | - |

Following initial antimicrobial activity screening, the metabolites of endophytes in DMSO extract (10 mg/mL) of selected fungal crude extracts showing promising bioactive activities against test organisms were further evaluated (Table 4). The endophytic fungal isolates PAL-4 and PAL-5 showed inhibition zones of 5.1 ± 0.2 mm and 6.0 ± 0.1 mm against

*B. subtilis* only. The extracts from isolates PAR-8 and PAS-7, in contrast, demonstrated inhibitory activity against all three bacterial strains tested. The fungal endophytic isolate PAR-14 demonstrated the largest zone of inhibition of $8.2 \pm 0.2$ mm against *E. coli* and $14.2 \pm 0.2$ mm against *S. aureus* compared to the ampicillin control with inhibition zones of $16 \pm 0.2$ mm and $16.3 \pm 0.2$ mm, respectively. Among the endophyte isolates tested against *B. subtilis*, only DMSO extracts from isolate PAS-07 showed the largest zone of inhibition of $13.0 \pm 0.1$ mm, with the ampicillin control showing a $21.1 \pm 0.2$ mm zone of inhibition. None of the isolates showed significant antimicrobial activity against fungal stains tested at the concentration of 10 mg/mL in DMSO.

**Table 4.** Antimicrobial activity of the DMSO extracts of endophytic fungi against selected microbial pathogens (the numbers are the diameters of the zones of inhibition in mm; mean $\pm$ SD; $n = 3$); Positive control 1: Ampicillin sodium (100 µg/mL); Positive control 2: fluconazole (30 µg/mL); Negative control: 10% DMSO).

| Isolates | E. coli | S. aureus | B. subtilis | C. albicans | A. fumigatous |
|---|---|---|---|---|---|
| PAL04 | - | - | $5.1 \pm 0.2$ | - | - |
| PAL05 | - | - | $6.0 \pm 0.1$ | - | - |
| PAL23 | $4.2 \pm 0.1$ | $2.2 \pm 0.2$ | - | - | - |
| PAR08 | $8.1 \pm 0.2$ | $11 \pm 0.2$ | $8.1 \pm 0.1$ | - | - |
| PAR14 | $8.2 \pm 0.2$ | $14.2 \pm 0.2$ | - | - | - |
| PAR16 | $7.2 \pm 0.2$ | $13.2 \pm 0.1$ | - | - | - |
| PAS04 | - | $10.1 \pm 0.2$ | - | - | - |
| PAS07 | $8 \pm 0.2$ | $11.3 \pm 0.3$ | $13.0 \pm 0.1$ | - | - |
| PAS21 | - | $11.1 \pm 0.1$ | - | - | - |
| Positive control 1 | $16 \pm 0.2$ | $16.3 \pm 0.2$ | $21.1 \pm 0.2$ | - | - |
| Positive control 2 | - | - | - | $24.1 \pm 0.3$ | $17.4 \pm 0.1$ |
| Negative control | - | - | - | - | - |

### 3.3. Molecular Characterization

The eight isolates showing strong antimicrobial activity (>20 mm) from fungal crude extracts were chosen to sequence the internal transcribed spacer region (ITS) of ribosomal DNA. Sequenced data revealed, as shown in Table 5, that *Fusarium solani, F. oxysporium*, and *F. proliferatum* are the main endophytic fungi that are responsible for antimicrobial activity. All the sequenced fungal isolates were *Fusarium*, which belongs to the Phylum Ascomycota, Class Sordariomycetes, and Order Hypocreales. Phylogenetic tree analysis demonstrated that *Fusarium oxysporium* OR055973.1 formed the group with *F. oxysporium* OR055970.1 with a strong bootstrap support of 93%. Similarly, *F. solani* OR055975.1 and *F. solani* OR055974.1 form a group with a bootstrap value of 64%. Results indicate that all the sequenced isolates belong to the same genus, *Fusarium*.

**Table 5.** Sequencing results of selected isolates based on their antimicrobial activities.

| Sample | Tissue | ITS Identity | Classification | Accession No. |
|---|---|---|---|---|
| PAL05 | Leaf | 99.62% | *Fusarium solani* | OR054275.1 |
| PAL23 | Leaf | 92.68% | *Fusarium oxysporum* | OR055969.1 |
| PAS04 | Stem | 99.62% | *Fusarium oxysporum* | OR055970.1 |
| PAS13 | Stem | 100% | *Fusarium solani* | OR055971.1 |
| PAR03 | Root | 99.82% | *Fusarium solani* | OR055974.1 |
| PAR06 | Root | 100% | *Fusarium oxysporum* | OR055973.1 |
| PAR13 | Root | 99.64% | *Fusarium solani* | OR055975.1 |
| PAR16 | Root | 100% | *Fusarium oxysporum* | OR055976.1 |

### 3.4. Statistical Analysis

The result of the two-way Manova test for maximum value showed that there is a statistically significant difference between the means of the dependent variables: concentration of extract tested (M 100, M 150, and M 200) and the two factors (factor 1: *Bacterial* spp. and factor 2: *Endophytic fungal* spp.), i.e., $p \leq 0.01$. The individual ANOVA tests and Wilk's Lambda test for the dependent variables M 100, M 150, and M 200 also demonstrated that both factors are statistically significant ($p \leq 0.01$).

## 4. Discussion

The endophytic fungus possesses unique and diverse antimicrobial activity [31]. The diversity of fungi is unknown; the number of endophytic fungi is estimated to be approximately one million based on the 6:1 ratio of fungal–plant species [32]. However, in tropical and subtropical regions, the ratio could be up to five times higher [33]. In this study, endophytic fungi isolated from the medicinal plant *Petiveria alliacea* were evaluated to investigate the potential for producing novel bioactive compounds.

The medicinal herb *Petiveria alliacea* has gotten a lot of attention in recent years, and its extracts possess many metabolites that have the potential to kill or inhibit many types of disease-causing pathogens [34,35]. Previous studies have shown that the benzyl-containing thiosulfinates from *P. alliacea* exhibited the broadest spectrum of antimicrobial activity against bacteria and fungi [36]. The present study was initiated to understand the association of fungal endophytes with *Petiveria alliacea* in South Florida, as there was no such research conducted previously.

Many studies have been undertaken and established that endophytic fungal crude extracts have antibacterial properties [34,35]. These results might be attributed to the fact that the extract may contain a high concentration of bioactive metabolites. The fungal extract that showed very low antimicrobial activity in the bioassay might have contained active bioactive metabolites in low or very low concentrations or might have yielded more active compounds if it was further purified [34]. Various factors might influence the qualitative and quantitative aspects of bioactive metabolite production by endophytic fungi under laboratory conditions. The spectrum and degree of inhibitory activity of endophytic fungal metabolites can also be affected by the type of solvent used [37].

The fungal endophytes isolated from *Petiveria alliacea* were tested against *E. coli*, *S. aureus*, *B. subtilis*, *C. albicans*, and *A. fumigatous* to test their production of antimicrobial compounds by using a dual culture assay. These initial results hinted that those isolates might produce bioactive compounds. They were then tested using agar well diffusion assay techniques (Figure 4). The zone of inhibition calculations was calculated based on agar well diffusion techniques as it was more reliable than the dual culture bioassay technique.

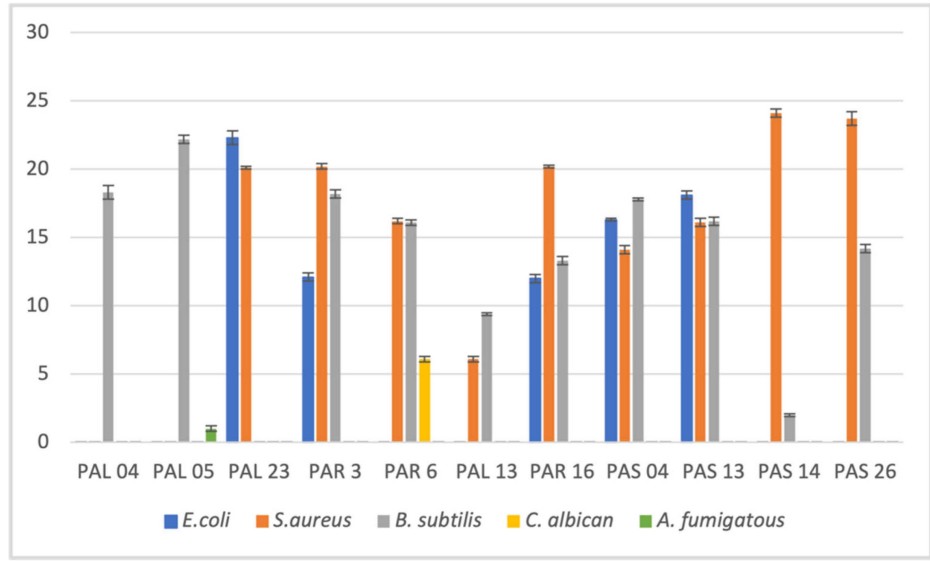

**Figure 4.** Mean zone of inhibition in mm of the crude extracts (*y*-axis) of the fungal endophyte isolates against bacterial and fungal strains (*x*-axis) using 100 µL.

Sequencing data from this study demonstrated that the endophytic fungal strain sequenced is a species of *Fusarium* (Figure 5). Endophytic strains of *Fusarium* are known for producing many different types of useful bioactive compounds. The antimicrobial drug PTOX (podophyllotoxin) was produced by the endophytes *F. oxysporum* isolated from

*Juniperus recurve* [38]. Another natural product produced by the endophytic fungi *Fusarium* is Taxol, which is the first billion-dollar drug produced by *F. proliferatum* from the Taxol plant [39,40]. Crude extracts of endophyte *F. equiseti* isolated from *Mikania cordata* showed a significant broad-spectrum of antimicrobial activity against *Bacillus cereus*, *Staphylococcus aureus*, *Escherichia coli*, and *Pseudomonas aeruginosa* [41]. A polyketide fusaequisin A extracted from *F. equiseti* was found to be inhibitory to *S. aureus* and *P. aeruginosa* [42]. *Fusarium* spp. are also known to produce antimicrobial compounds such as PTOX, beauvericin, and subglutinol A and B [22].

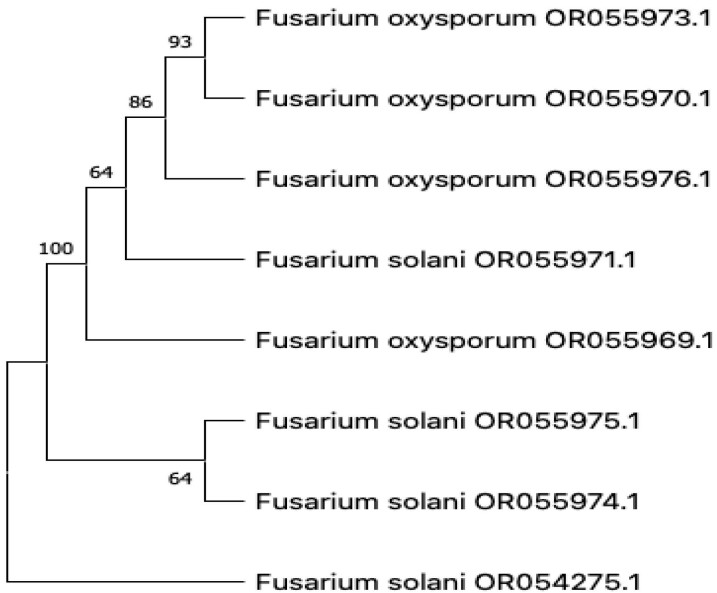

**Figure 5.** Phylogenetic (Neighbor-joining) tree based on sequence data from ITS gene. The value on each branch is the percentage of bootstrap replications supporting the branch.

An additional antibacterial substance produced by endophytes *Fusarium* spp. identified from medicinal plants are 2-methyl butyraldehyde-substituted α-pyrone, beauvericin, and subglutinol A and B [27,43,44]. Endophytic fungi such as *F. solani*, *F. oxysporium*, and *F. proliferatum* have the potential to produce different types of bioactive metabolites and are supported by studies of other medicinal plants all around the world [45–49].

The antimicrobial activities of different species of *Fusarium* have been attributed to the production of bioactive secondary metabolites (SMs) such as terpenoids, polyketides, steroids, quinones, flavonoids, alkaloids, peptides, and these metabolites [50–52]. *Fusarium* sp. has been reported to produce anti-Gram-negative bacterial SMs, anti-Gram-positive bacterial SMs, both anti-Gram-positive and anti-Gram-negative bacterial SMs, antifungal SMs, and both antibacterial and antifungal SMs [53].

The metabolites produced by endophytic fungi isolated from medicinal plants have antibacterial, antifungal, anticancer, and antioxidant properties, and it might provide opportunities to explore more endophytic fungi for discoveries in the field of antimicrobials [52,54]. However, more studies are needed to address the dynamical changes of endophytic communities [54] and unculturable fungal communities [55].

## 5. Conclusions

The present data showed that *P. alliacea* root contains the highest diversity of fungal endophytes. This study demonstrated that 37% of morphologically different fungal isolates were isolated from the root, whereas 33% were from the leaf and 30% were from the stem sample. Isolates that have the strongest antimicrobial activities in the three tissue samples also have similar genetic makeup as per the results from the fungal isolates that we sequenced.

This is the first study to explore endophytic fungi of *Petiveria alliacea* and evaluate their potential in vitro antimicrobial activities. Antimicrobial activity was tested on a total of 30 morphologically diverse fungal isolates. The isolates with a good zone of inhibition were sequenced to determine their identity. The crude extracts of endophytic fungal isolates contain promising antimicrobial compounds and deserve further purification and characterization for their chemical constituents. Future studies need to investigate all the culturable isolates, as well as unculturable isolates, using molecular methods for their identification. Expanding the testing of identified isolates for more pathogens and drug-resistant strains would help to solve the global problem of antibiotic resistance.

**Author Contributions:** Conceptualization: K.J., K.G.S., T.A., Y.-C.T.-D. and G.K.; Methodology: G.K., K.J., K.G.S., T.A. and Y.-C.T.-D.; Lab work: G.K. completed the lab work; Funding acquisition: K.J., Y.-C.T.-D. and K.G.S.; Resources managements: K.J., K.G.S., Y.-C.T.-D. and T.A.; Data analysis and validation: G.K., K.G.S., T.A., Y.-C.T.-D. and K.J.; Original draft preparation: G.K.; Manuscript writing—review and editing: K.J., Y.-C.T.-D., T.A. and K.G.S. reviewed and finalized manuscript. All authors have read and agreed to the published version of the manuscript.

**Funding:** This study was supported by the Florida International University, College of Arts, Sciences and Education. This project was also supported by Florida International University through a graduate teaching assistantship as well as from the Department of Earth and Environment at Florida International University.

**Institutional Review Board Statement:** Not applicable.

**Informed Consent Statement:** Not applicable.

**Data Availability Statement:** The original contributions for this study are included in the article, and further inquiries can be directed to the corresponding author.

**Acknowledgments:** We would like to thank Diego Salazar Amoretti for the technical resources and equipment used at the plant chemical ecology lab at Florida International University. Also, we would like to thank Amir Khoddamzadeh for providing valuable suggestions.

**Conflicts of Interest:** While conducting the research, all the authors declare that they have no conflict of interest. The final decision on the manuscript preparation and submission of the findings for publication had been reached by all the authors.

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
