# Peer review of "Isolation and Characterization of Fungal Endophytes from Petiveria alliacea and Their Antimicrobial Activities in South Florida"

_2036-7481, doi:10.3390/microbiolres14030100_

Round 1

Reviewer 1 Report

The authors revealed antimicrobial production of endophytic fungi isolated from Petiveria alliaces. The findings would be contributed to investigate drug resistance pathogen. So, this manuscript is valuable to publish in Journal of Fungi. But I would like to recommend to revise the manuscript to clear the data to understandable easily before publication.

Here, I would like to pint out the revision points.

Line 120: 0.5cm >> 0.5 cm (insert space before cm)

Line 203: 11>>Eleven

Line 236: Table2

This table is too complicated. So, I can not distinguish each column.  Please inset space between column.

Line306: Figure 4.

I can not find what meaning. Is the figure showing colony inhibition diameter?

Please insert explanation of Y axis.

Author Response

Response to reviewers’ comments and suggestions

Reviewer 1:

The authors revealed antimicrobial production of endophytic fungi isolated from Petiveria alliaces. The findings would be contributed to investigate drug resistance pathogen. So, this manuscript is valuable to publish in Journal of Fungi. But I would like to recommend revising the manuscript to clear the data to understandable easily before publication.

Here, I would like to point out the revision points.

We greatly appreciate all your comments, concerns, and suggestions. Thank you very much for all your feedback.

(1) Line 120: 0.5cm >> 0.5 cm (insert space before cm)

Done

(2) Line 203: 11>>Eleven

Done

(3) Line 236: Table2

Done

(4) This table is too complicated. So, I cannot distinguish each column.  Please insert space between columns.

Thank you for your suggestion. We have inserted the space between the columns.

(5) Line306: Figure 4.

Done

(6) I cannot find what meaning. Is the figure showing colony inhibition diameter?

Please insert explanation of Y axis.

We have inserted axis titles to the figure 4.

Reviewer 2 Report

comments need to responses 

1-  why used  Positive control 1: Ampicillin sodium (100 µg/ml),  however, the tested organisms were fungi in Table 3

2- Figure 2. Antibacterial activity of the extracts of fungal endophytes fungi against selected bacterial strains was poor, where the clear zones in some plates were confused. if the author minimize the concentration of extract may give suitable clear zones 

3-  all scientific names must be italic 

4- line 313 That had suggested that those isolates might produce bioactive compounds. Therefore HPLC or GC-MS or separates the active constituents must be  need

5-  line 325 acillus cereus, Staphylococcus aureus, Escherichia coli, and Pseudomonas aeruginosa  must be acillus cereus, S. aureus, E. coli, and Pseudomonas aeruginosa 

6. discuss the possible mechanisms of antimicrobial activity of isolated fungi

7- why cite the reference in the conclusion such as (Kaul et al., 350 2013; Mishra et al., 2012). please  focus on your results

8-  the title must express the performed works, therefore must be changed   to Isolation and characterization of fungal endophytes from Petiveria alliacea and their antimicrobial activities in South Florida

9-  added the novelty of the work in the introduction and compare its and other previous study.

10-  how to identify the collected plant  Pativeria alliacea , must add the taxonomist that helps for identification

11- is you used one medium (PDA) for the isolation and identification of isolates

12-  provide references in each part of the methods

13- line 152 (Escherichia coli ATTC 25902),Gram-positive (Staphylococcus aureus ATTC 14775, Bacillus subtilis NRRL 5109), Candida albicans ATTC 10231), and multicellular filamentous fungus (Aspergillus fumigatus NRRL 5109)  must italic

Moderate editing of English language required

Author Response

Response to reviewers’ comments and suggestions

Reviewer 2:

We greatly appreciate all your comments, concerns, and suggestions. Thank you very much for all your feedback.

(1) why used Positive control 1: Ampicillin sodium (100 µg/ml), however, the tested organisms were fungi in Table 3

Thank you very much for pointing this out. We have removed Ampicillin sodium (100 µg/ml) as the positive control.

(2) Figure 2. Antibacterial activity of the extracts of fungal endophytes fungi against selected bacterial strains was poor, where the clear zones in some plates were confused. if the author minimizes the concentration of extract may give suitable clear zones 

Thanks for your comment. The images in the Figure 2 were included to show a range of antibacterial inhibition activity of fungal crude extracts observed. We did include quantitative values of inhibition zones in tables 2 and 4.

(3) all scientific names must be italic 

Thank you. We made the change throughout the manuscript.

(4) line 313 That had suggested that those isolates might produce bioactive compounds. Therefore, HPLC or GC-MS or separates the active constituents must be need.

Thank you for your suggestion. This is an ongoing project, and we are working on the HPLC / GC-MS based fractionation, purification, and chemical characterization of active constituents.

(5) line 325 Bacillus cereus, Staphylococcus aureus, Escherichia coli, and Pseudomonas aeruginosa must be Bacillus cereus, S. aureus, E. coli, and Pseudomonas aeruginosa 

Thank you. We made the change throughout the manuscript.

(6) discuss the possible mechanisms of antimicrobial activity of isolated fungi

Thank you for your suggestion. We have updated the possible mechanism of antimicrobial activities in the manuscript in line 340 as:

The antimicrobial activities of different species of Fusarium have been attributed to production of bioactive secondary metabolites (SMs) such as terpenoids, polyketides, steroids, quinones, flavonoids, alkaloids, and peptides and these metabolites [50–52]. Fusarium sp. have been reported to produce anti-gram-negative bacterial SMs, anti-gram-positive bacterial SMs, both Anti-Gram-Positive and anti-gram-negative bacterial SMs, anti-fungal SMs, and both anti-bacterial and anti-fungal SMs [53].

(7) why cite the reference in the conclusion such as (Kaul et al., 350 2013; Mishra et al., 2012). please focus on your results

Revised accordingly.

(8) the title must express the performed works, therefore must be changed   to Isolation and characterization of fungal endophytes from Petiveria alliacea and their antimicrobial activities in South Florida

Thank you for your suggestion. Revised accordingly.

(9) added the novelty of the work in the introduction and compare its and other previous study.

We have added a paragraph to the Introduction:

The endophytic fungal communities of Petiveria alliacea from South Florida and their bioactive potential have not been examined before. The purpose of this study was to investigate endophytic fungal microbiomes of native medicinal plant Petiveria alliacea in South Florida as source of novel antimicrobials to combat the problem of emerging new biothreats and antimicrobial resistance.

(10) how to identify the collected plant Pativeria alliacea, must add the taxonomist that helps for identification

We have added the taxonomist in the manuscript.

(11) is you used one medium (PDA) for the isolation and identification of isolates

Potato Dextrose Agar is composed of dehydrated Potato Infusion and Dextrose that encourage flourishing fungal growth. Besides that, the nutritionally rich base (potato infusion) encourages mold sporulation and pigment production in most fungi.

The potato dextrose agar (PDA) amended with antibacterial antibiotics is the most frequently used medium for recovery of fungal endophytes (Dos Reis et al. 2022). In addition, compared to other fungal media the greatest number of endophytic fungi were isolated on the PDA (An et al. 2020). For these reasons, we chose PDA as the medium of choice for this project.

Dos Reis JBA, Lorenzi AS, do Vale HMM. Methods used for the study of endophytic fungi: a review on methodologies and challenges, and associated tips. Arch Microbiol. 2022 Oct 20;204(11):675. doi: 10.1007/s00203-022-03283-0. PMID: 36264513; PMCID: PMC9584250.

An C, Ma S, Shi X, Xue W, Liu C, Ding H. Isolation, diversity, and antimicrobial activity of fungal endophytes from Rohdea chinensis (Baker) N.Tanaka (synonym Tupistra chinensis Baker) of Qinling Mountains, China. PeerJ. 2020 Jun 17;8:e9342. doi: 10.7717/peerj.9342. PMID: 32596051; PMCID: PMC7305772.

(12) provide references in each part of the methods

Thank you for your suggestion. We have included references in each part of the methodology.

(13) line 152 (Escherichia coli ATTC 25902), Gram-positive (Staphylococcus aureus ATTC 14775, Bacillus subtilis NRRL 5109), Candida albicans ATTC 10231), and multicellular filamentous fungus (Aspergillus fumigatus NRRL 5109)  must italic

Thank you. We made the change throughout the manuscript.

(14) Comments on the Quality of English Language

Moderate editing of English language required.

Thank you. We have checked the English and corrected grammatical errors.

Round 2

Reviewer 2 Report

Accept in present form